# Use of Danish National Somatic Cell Count Data to Assess the Need for Dry-Off Treatment in Holstein Dairy Cattle

**DOI:** 10.3390/ani13152523

**Published:** 2023-08-04

**Authors:** Maj Beldring Henningsen, Matt Denwood, Carsten Thure Kirkeby, Søren Saxmose Nielsen

**Affiliations:** Animal Welfare and Disease Control, Department of Veterinary and Animal Sciences, Faculty of Health and Medical Sciences, University of Copenhagen, 1870 Frederiksberg, Denmark; md@sund.ku.dk (M.D.); saxmose@sund.ku.dk (S.S.N.)

**Keywords:** PCR testing, mastitis, wilmink function

## Abstract

**Simple Summary:**

This scientific study aimed to compare somatic cell count (SCC) curves throughout lactation in dairy cows according to PCR test results for four major intramammary infection (IMI) pathogens. Data from 133,877 Holstein cows in Danish conventional dairy herds were analysed using a nonlinear mixed-effects model with a modified four-parameter Wilmink function. The PCR tests were conducted before dry-off to determine animals eligible for selective dry cow therapy. The findings revealed that cows testing positive for IMI pathogens had higher mean SCC across all parity groups and lactations. Using SCC data fitted to the entire lactation allowed for quantifying overall differences in SCC curves.

**Abstract:**

In Denmark, PCR testing of dairy cattle is commonly used to select animals for the antibacterial treatment of intramammary infection (IMI) during the dry-off period. IMI is associated with a high somatic cell count (SCC), routinely recorded for milk quality control for most commercial dairy herds. This study aimed to compare SCC curves over the lactation among dairy cows with positive vs. negative PCR test results for four major IMI pathogens. Data from 133,877 PCR-tested Holstein cows from 1364 Danish conventional dairy herds were used to fit a nonlinear mixed-effects model using a modified four-parameter Wilmink function. We stratified the data into first, second, third or fourth and later parity and fitted Wilmink curves to all SCC observations between 6 and 305 days in milk. The PCR tests were taken before dry-off at the end of the lactation to investigate which animals qualified for selective dry cow therapy. A PCR Ct-value of 37 and below was used to determine if an animal was PCR positive for any of the following IMI pathogens: *Staphylococcus aureus*, *Streptococcus agalactiae*, *Str. dysgalactiae* and *Str. uberis*. Our findings showed that mean SCC curve fits were higher for PCR-positive animals in all four parity groups and across lactations. The use of SCC data fitted to the entire lactation for multiple lactations enabled quantification of overall differences in SCC curves between cattle with and without detected IMI, adjusted for parity group and stage of lactation. These findings are relevant to the use of SCC to support treatment decisions.

## 1. Introduction

Increased levels of antibacterial resistance present a significant health risk to humans and animals [1]. Food producers are strongly encouraged to reduce antibacterial usage without affecting animal welfare and food safety, and the links between public health, food safety, food security, and animal health & welfare are also central in the EU Animal Health Law from 2016 [2], which came into force in April 2021. Thus, reduced usage of antibacterials needs to be balanced against the disadvantages of untreated disease in terms of animal health and productivity.

The main use of antibacterials in dairy cattle is for treating intramammary infection (IMI) [3]. IMI is common in dairy production [4] and has a damaging effect both on milk production and animal welfare. It is, therefore, essential to keep the number of IMI cases low [5,6]. In the past, a systematic treatment known as blanket dry cow therapy (DCT) was employed to reduce IMI, meaning all animals were prophylactically treated with antibiotics. Dairy producers used blanket DCT during dry-off to reduce IMI and subclinical and clinical mastitis in the following lactation [7]. However, to limit the prophylactic use of antibacterial, many countries, including Denmark, have banned blanket DCT, replacing it with selective DCT (SDCT) [8,9]. In SDCT, animals are selected for antibacterial treatment to ensure animal health and productivity following dry-off, leaving animals that do not need antibacterial treatment untreated [10].

In Denmark, milk samples’ bacterial culture or PCR testing is used to detect the presence of IMI pathogens for SDCT. The most common pathogens include coagulase-negative *staphylococci* (CNS), *Staphylococcus aureus*, *Streptococcus uberis*, *Str. dysgalactiae*, *Str. agalactiae*, *Corynebacterium bovis*, and *Escherichia coli* [11,12,13]. Treatment is permitted in Denmark as long as *Staphylococcus aureus*, *Str is present. agalactiae*, *Str. dysgalactiae* or *Str. uberis* has been confirmed by a positive PCR test or bacteriological culture [9]. However, the sensitivity of bacteriological culture is not high [14], and expensive PCR tests can give false positive results based on routinely recorded milk samples [15,16].

Since IMI is typically associated with increased somatic cell count (SCC) [17,18], routine recording of milk quality offers another, and potentially less expensive, approach to the detection of IMI [19,20,21,22]. In several countries, including Denmark, SCC exceeding 200,000 cells/ml usually indicates IMI [23,24], but typical SCC varies depending on which pathogens are present [25,26]. Other factors affecting SCC levels include days in milk (DIM), parity, breed, and milk yield [11,27].

As of May 2021, Danish dairy cows can be selected for DCT not only based on a diagnostic test confirming the presence of an IMI pathogen but also if their SCC level has been measured above 200,000 cells/ml at least twice in the last four months before treatment at dry-off [28]. However, this general threshold ignores important factors affecting SCC. For example, the SCC varies greatly by parity and stage of lactation [27]. By fitting a lactation-type model, such as the Wilmink-curve, to SCC measurement [29] and including the additional factors, a better prediction for IMI is potentially possible. The results can be used to monitor dairy cattle welfare [25] and to support treatment decisions so that animals are not treated unnecessarily.

The objective of this study was to characterise the association between PCR test results at dry-off and SCC curves before dry-off in a population-based context using registry data and precisely to fit an SCC curve model that can be used to support treatment decisions for four of the significant mastitis pathogens.

## 2. Materials and Methods

In this section, we describe the details of our analysis. Briefly, our approach was to make parity-specific comparisons of somatic cell count lactation curves between PCR-positive (to one or more of four major mastitis pathogens) and PCR-negative cows.

### 2.1. Data

This was an observational retrospective registry study. The data included routinely recorded data from all Danish dairy herds, monitored using 11 milk recordings per year, which is the standard in the Danish milk recording scheme. Data from January 2010 to May 2020, including SCC recordings for each test date, calving dates, herd type and PCR laboratory results, were collected from the Danish Cattle Database (DCD, SEGES P/S, Aarhus N, Denmark). Eurofins Stein’s Laboratory (Vejen, Denmark) recorded the SCC and provided the PCR tests for the DCD, including 15 mastitis pathogens and one antibiotic resistance gene, staphylococcal beta-lactamase [30], using the PathoProof 16 PCR kit (Thermo Fisher Scientific, Waltham, MA, USA), as described elsewhere [31]. The target population was all Danish dairy cattle in conventional production.

Data cleaning was carried out as follows. Holstein cattle, which comprise the majority (65%) of Danish dairy cattle herds [32,33], were included. The reason for excluding other breeds was to keep modelling as simple as possible without having to take breed-specific SCC patterns into account. Organic herds were excluded to ensure the herds followed the exact legal requirements for testing and treatment at dry-off, leaving only conventional dairy herds. Animals were grouped into Parity 1, Parity 2, Parity 3, or Parity >3. DIM was restricted to the range of 6–305 days because SCC levels tend to be unstable over the first five days [27], and the Wilmink-function for lactation curves on milk covers up to 305 days [29]. SCC recordings with values of zero were treated as errors and deleted. SCC, measured as 1000 cells/ml, was log-transformed to improve the normality of model residuals. Lastly, only data from animals with a PCR test in the dry-off period after the lactation phase were included, and the data were then grouped into PCR positive or negative. Samples testing positive for at least one of the four IMI pathogens of interest (*St. aureus*, *Str. agalactiae*, *Str. Dysgalactiae*, and *Str. uberis*) were categorised as positive, and the remainder negative, regardless of positive results for any of the 11 remaining pathogens. The threshold for a positive PCR test was set at a Ct-value of 37, following the official Danish guidelines for treatment at dry-off [9,34]. A flowchart of the data-cleaning process is presented in Figure 1.

### 2.2. Descriptive Analysis

Descriptive statistics were carried out using the tidyverse package in R [35,36]. The variables of interest in the final data included: Herd identification number, Parity group, PCR result, SCC, and DIM. Summary statistics had the SCC geometric mean, median, minimum, 1st and 3rd quartile and maximum, stratified by Parity group and PCR result.

### 2.3. Somatic Cell Count

The log-transformed SCCs were stratified by parity group (parities 1, 2, 3, >3) and PCR result (positive/negative) before being fitted to stratum-specific nonlinear mixed-effects (NLME) models relating DIM to log SCC using a modified four-parameter Wilmink-function [27,29]. The original Wilmink-function for milk yield generally describes the lactation curve, with an initial rise to a maximum followed by a steady and slow decrease throughout the lactation phase. Thus, the inverted function describes the SCC level throughout the lactation, starting with a quick drop before meeting a minimum, followed by a steady rise:log(SCC)*_j,t_* = *a_j,t_* + *b_j,t_* × DIM + exp(−exp(*c_j,t_*) × DIM) × *d_j,t_*,(1)
where the *a*, *b*, *c* and *d* are the four Wilmink-parameters describing the SCC curve. Note that the Wilmink-curve is fitted separately to all eight combinations of parity group *j* and test result *t* (denoted by subscripts *j* and *t*).

Random slopes of the herd were fitted to each of the four Wilmink-parameters to allow for variation in the SCC curve between herds, but variation between individual animals in the same herd was ignored. We note that the random slope for the parameter *a* (effectively the intercept) is equivalent to a random intercept, but we refer to each as random as slopes for consistency. The data from each stratum (*j* and *t*) were fitted to the model separately using the nlme function in R, which maximises the log-likelihood of the model using a normal response for the data [37]. To reduce the computational burden of finding a global optimum for the mixed-effects model, the data were first fitted to a simpler nonlinear least square (NLS) model (i.e., without the random slope of the herd), using the nls.multistart function in R [38,39]. The coefficients output by the NLS model were then used as initial parameters for the NLME fit to reduce the time to convergence.

Estimated Wilmink-parameters for each herd from the eight NLME fits were retrieved, and logSCC curves were constructed using the mean of the parameters for each parity and PCR group, resulting in eight estimated SCC curves. Summary statistics were then calculated for each of the eight curves, including the minimum SCC value and the rate of change of SCC from DIM 100 to 150, which indicates the slope of the approximated linear increase after the minimum values. The distribution of the estimated NLME Wilmink-parameters was then plotted using the estimated cumulative distribution function (ECDF), together with the 95% confidence interval (CI) of the parameters obtained from the model fit, thereby visualising the different rates between PCR positive and negative for each parity group.

The fit of the NLME models was assessed by visual evaluation of the residuals. Quantile-Quantile (Q-Q) plots were generated to determine whether the residuals were normally distributed. Fitted vs. residuals plots were created to assess the difference between the predicted and actual values.

## 3. Results

### 3.1. Descriptive Analysis

A total of 2,564,032 unique animals and 3940 herds were represented in the raw DCD production data for the study period 2010–2020. After removing SCC observations with no information and filtering for eligible data in accordance with the inclusion criteria (Figure 1), 133,948 animals and 1395 herds were included in the analysis, with some of the animals being represented in more than one parity. SCC variables for each of the data groups are summarised in Table 1. Increases in the mean and median as Parity increased can be observed. Overall, SCC levels in the PCR-positive groups were higher than in the PCR-negative groups.

### 3.2. Somatic Cell Count Modelling

Fitting the NLME resulted in eight separate model fits (each with four Wilmink-parameter estimates plus random slopes for each herd) for each data subset, as stratified by parity and PCR result. Assessment of model fit (Figure 2) revealed large discrepancies relative to the assumed normal distribution, so interpretation of presented *p*-values should be done with care. Summary statistics for the estimated population-level Wilmink-parameters from the NLME model for each of these strata are shown in Table 2. The estimated *d* parameter for PCR positive and group Parity >3 had a notably higher standard deviation (SD) of 1.91 than the other estimated strata.

In addition to population-level estimates for each parameter, our use of random slopes for herd allows us to examine the distribution of conditional modes of the random effect estimates. Figure 3 shows an empirical cumulative distribution function (ECDF) for the four estimated Wilmink-parameters for the two data strata involving Parity 2. A wider between-herd variation in the estimated parameter *c* and *d* values was observed in PCR positive group, resulting in overlapping ECDF curves for these two parameters. This means some PCR-positive herds had lower conditional mode estimates than some PCR-negative herds. As for *a* and *b*, more similar but separated ranges were observed in the PCR negative and positive groups.

The estimated logSCC curves for each parity and PCR group by DIM are shown in Figure 4. The average overall SCC level throughout the lactation found in the PCR-positive group was higher than in the PCR-negative group. For both groups, it was found that the SCC level increases with the parity group on average.

The summary statistics retrieved from each model are shown in Table 3. The lowest estimated minimum values for SCC are in Parity group 2 in the PCR negative group, while the highest are in the Parity group >3 PCR positive group. For each parity group, a higher estimated minimum value of SCC was found for the PCR-positive group, and the estimated minimum SCC values increased with parity. The time, measured in DIM, corresponding to the minimum SCC, also differed between the PCR and parity groups. DIM for the minimum SCC was higher in the PCR negative than in the PCR positive groups. DIM for minimum SCC decreased with increasing parity. The slope of the estimated logSCC curves, calculated from DIM 100 to 150, also depended on parity and PCR group, as the PCR negative groups had a slower increase than the PCR positive group, with the rate increase with higher parity groups.

## 4. Discussion

We present an approach to fit SCC curves using a modified Wilmink-function. The SCC curves were compared between PCR-positive and PCR-negative cattle for one or more of the four most common IMI pathogens affecting Danish Dairy herds in the period before dry-off. This is the first study using an NLME model to fit routinely recorded SCCs to the entire lactation from all eligible Danish dairy herds, i.e., 1395 herds with routine SCC measurements. All summary statistics shown are population averages, although variation in parameters between herds was accounted for by our model. Variation between animals in the same herd was not accounted for. We also note that the model residuals showed some deviation from being normally distributed, which can be expected to impact the reliability of confidence intervals and *p*-values. However, we believe that our extremely large sample size means that the impact of this departure from normality on the effect estimates that we present is minimal.

Our method allows patterns to be detected within recorded high-volume SCC data. However, as the quantity of recorded health data increases, the handling of these data in epidemiological studies must be considered. The use of large datasets with heterogeneous variation between subgroups and real-time analysis is likely to grow. A range of skills, tools, and methods are required to handle these data [40]. The data can include parameters that can affect the risk of IMI, such as herd size, treatment scheme, season of year, previous diseases, previous treatment, and prophylactic treatments [41,42,43]. Thus, for recorded high-volume SCC data from dairy herds, we suggest that a modified Wilmink-function fitted with NLME models is suitable, but it would also be interesting to fit SCC to other modified milk curve functions such as Wood’s function [44]. However, when assessing SCC by the end of the lactation, the Wilmink function is preferred since Wood’s original curve for milk goes to zero for DIM going to infinity.

Evaluation of the goodness-of-fit of the models also suggests that other functions and models will ideally be considered. The Q-Q plots in Figure 2 are less than ideal, with the residuals deviating from the straight line in a similar pattern in all the models. We see more deviation in the negative models, which could indicate that different functions may be needed to fit negative and positive tested animals, respectively. The lack of good fit is also seen in the density residual plots in Appendix A.

The NLME-model fitting resulted in four estimated parameters for each data stratum (parity group and PCR test result) presented in Table 2, along with estimates of the mean for each herd based on the conditional modes of the corresponding random effects. For the Parity group >3, the SD values for parameter *d* are higher than the *d* parameter in the other models. Animals in higher parities have been shown to receive more treatments in the lactation phase [45], which could generate greater variety in the estimated parameters. With *d* describing the initial drop towards the minimum on the logSCC curve, the high SD value can also be caused by fewer observations, as fewer data points are expected before than after the SCC minimum.

While it is known that IMI causes fluctuations in SCC [46,47], this is also the first study to examine how the predicted SCC level throughout lactation is affected by the presence of pathogens that are not disclosed or detected until the end of the lactation by a PCR test for SDCT. Our mean logSCC curves show that animals testing positive for one or more of the four main pathogens in Danish dairy herds at the end of the lactation have a higher SCC level throughout the lactation than animals that did not test positive. This result could be due to either a generally reduced immune function in the individual animal, resulting in elevated SCC, or to the causative pathogen found at the end of lactation. If the former is the case, the SCC curve model we described can be expanded to identify animals with reduced health. Furthermore, both DIM at minimum SCC values and the SCC increase on the estimated SCC curve are associated with the parity group and the PCR result. The relationship between parity and the rate of rise of SCC throughout lactation supports previous findings [27,48].

The differences in SCC levels between PCR positive and negative groups show an overall trend towards a higher SCC level for PCR positive strata, together with a higher DIM of the SCC minimum and a smaller increase in SCC after the minimum, which is mainly driven by the *b* parameter. This is even though animals in the PCR-negative group may have tested positive for other IMI pathogens than the four included in this study, and the possibility that this affected the SCC [49]. In addition, the potential for false positive tests, mainly due to contamination of the test, should not be neglected [50]. Thus, further research employing expanded datasets should ideally be conducted to identify the separate effect of each pathogen compared with animals with no recorded pathogens at dry-off. PCR thresholds, deviating from the default Ct-value cut-off at 37, could also be applied for each of the pathogens to establish the PCR threshold at which an elevated SCC level develops.

The patterns presented in this study support the Danish SDCT guidelines on using SCC for treatment decisions: the estimated mean of the PCR negative curves at the end of 305 days of lactation does not exceed an SCC level of 200,000 cells/ml (corresponding to 5.3 on the logSCC curves) in any of the parity groups, ensuring that there is no unnecessary use of antibacterials. However, for the Parity groups 1 and 2 that are PCR positive, the curve does not exceed 200,000 cells/ml at the end of lactation. Therefore, diagnostic tests and/or clinical findings may be more appropriate to support SDCT for younger animals.

Our findings also show that the typically recommended 200,000 cells/ml threshold for IMI [23,24] should be applied carefully, as SCC depends strongly on DIM and parity. All the presented curves start with SCC levels higher than 200,000 cells/ml, potentially indicating IMI, an unjustified treatment based on SCC levels alone in this phase. The length of this early phase may vary, but as all of the parity and PCR groups reached their SCC minimum levels within 40 DIM, it is reasonable to assume that the 200,000 cells/ml limit should not be applied before 40 DIM. Furthermore, some evidence suggests that a variable threshold (depending on both parity and DIM) would provide more consistent diagnostic accuracy than a single fixed threshold. However, this is on population averages, and as shown in Figure 3, there is a difference between herds, making it challenging to determine a globally applicable dynamic SCC threshold. At the cow level, significant increases in SCC from the previous lactation may be used as an alternative. Likewise, it can be expected that there will be differences between the individual animals, which we have not accounted for in this study, due to insufficient data points per animal.

Our study has several limitations. Firstly, we did not include the variation between animals in the same herd, as it is not feasible to include this number of random slopes in the model. This may have resulted in an artificial overestimation of the statistical significance (and under-estimation of 95% CI) for our parameter values, but these are not a central output of the study. The second limitation is the assumption that herd-level random slopes for each parameter can be adequately described by a normal distribution. We feel that this is justifiable due to the focus on conventional herds with Holstein cattle so that a large number of potential smaller differences between the relatively similar herds could be expected to be approximately normal due to the central limit theorem. However, we cannot exclude the possibility that a small number of critically important differences (e.g., robot vs. manual milking system) could create a multi-modal distribution that we have not accounted for. Thirdly, our models show a relatively poor fit to the data assessed by QQ plots. Due to the large number of observations, we believe that the output of the models is still qualitatively reasonable, but extreme care should be taken when interpreting precise quantitative comparisons. This has relevance for estimating the probability of positive vs. negative PCR tests at the end of lactation based on SCC values.

Certain limitations of research involving big data also need to be considered. We couldn’t verify that the application of the PCR tests (a selection criterion in the study) was random with respect to the animals’ SCC or mastitis status. PCR tests are applied voluntarily by a farmer for many potential reasons, but a large part will be to justify the treatment of animals that the farmer believes have a mastitis issue. Therefore, we would expect the SCC of PCR-negative animals in our study to be potentially quite different to the SCC of PCR -negative animals randomly selected for PCR testing. Given that approximately half of the available animals were excluded from our study due to a lack of PCR test results, this is a potentially large source of bias. However, we would, if anything, expect this bias to reduce the observed differences between PCR positive and negative animals, so we believe that our estimates of the differences are most likely conservative. There may, therefore, be some selection bias in the inclusion criteria for the study. Some animals may be tested because of suspicions of IMI, but we have no reason to believe this is the case in most herds, as all animals are usually tested as part of a herd strategy. Another limitation needing to be highlighted is that since the researcher does not collect registered data [51], important information on the herds may be missing in the study.

The main clinical implication of this study is that we need to move away from the traditional universal cut-off for dairy cattle in all lactations, e.g., 200,000 cells/ml. There are major differences across lactations and between lactations. However, once these differences are handled, it is clear that the approach taken here can be used not only for comparing PCR-positive and negative cows but also for other parameters, e.g., vaccination against mastitis pathogens. Furthermore, there is a need to develop models for cow-level assessment to enable cow-specific treatment options.

## 5. Conclusions

Our study demonstrates a method of estimating parameters for a Wilmink function based on routinely recorded SCC data. The study indicated that PCR positivity was associated with higher SCC values throughout the preceding lactation than in PCR-negative animals. Applying SCC modelling to support treatment decisions can help judicious and economical use of antimicrobial interventions, thereby improving animal health and mitigating the development of antimicrobial resistance.

## Figures and Tables

**Figure 1 animals-13-02523-f001:**
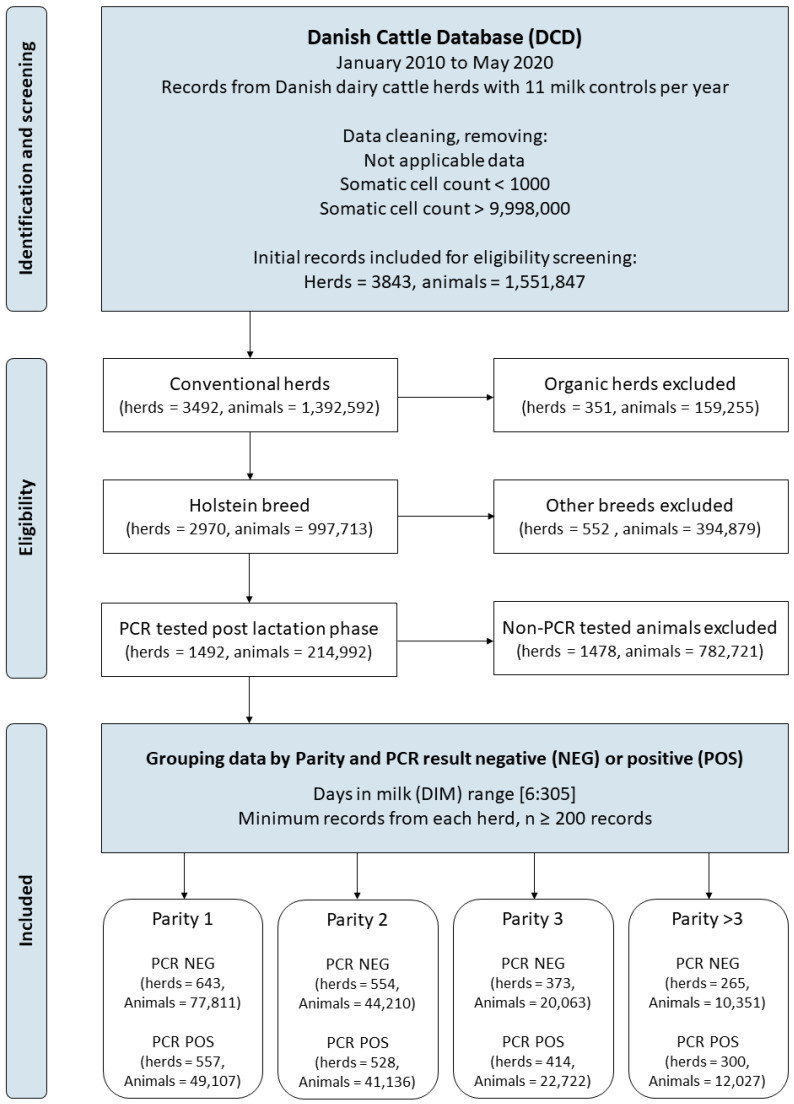
Flowchart showing the process for data cleaning and filtering of data for eligibility with an overview of the total number of herds and animals included in the modelling.

**Figure 2 animals-13-02523-f002:**
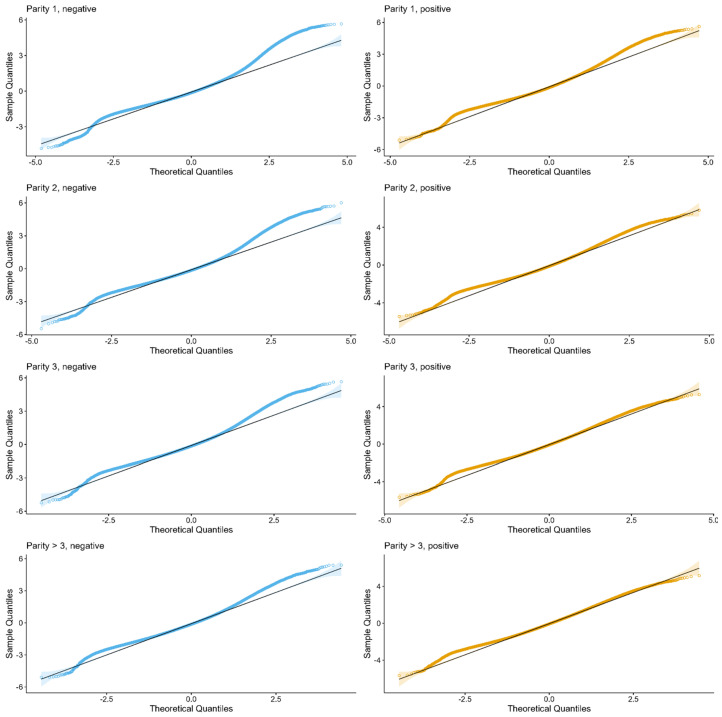
Q-Q plots of the residuals for the eight different models: to the left, with blue, PCR negative for Parities 1, 2, 3, and >3 and to the right, with orange, PCR positive for Parities 1, 2, 3, and >3.

**Figure 3 animals-13-02523-f003:**
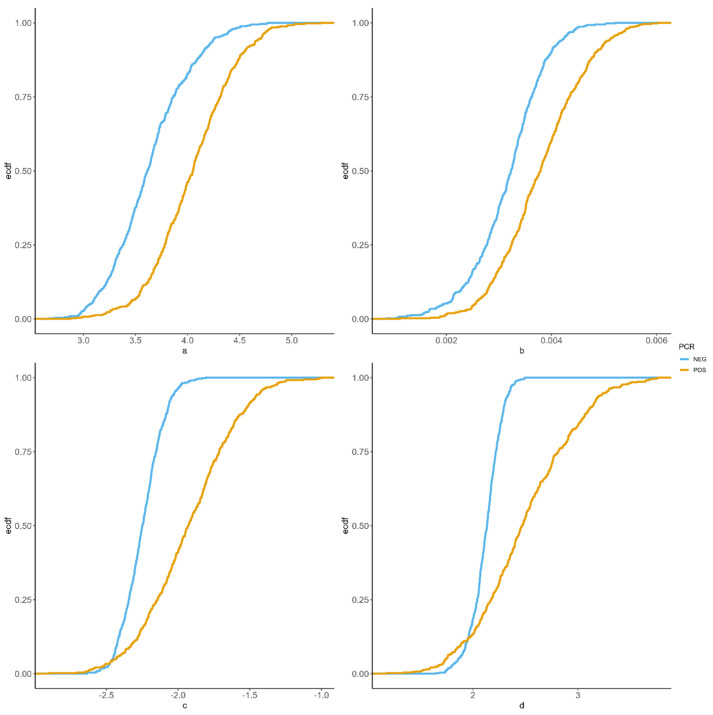
ECDF plots of the between-herd variation in parameter estimates (based on the conditional modes of random effects) for the four NLME estimated Wilmink-parameters, *a*, *b*, *c*, and *d*, for both PCR positive and negative strata of Parity 2 animals.

**Figure 4 animals-13-02523-f004:**
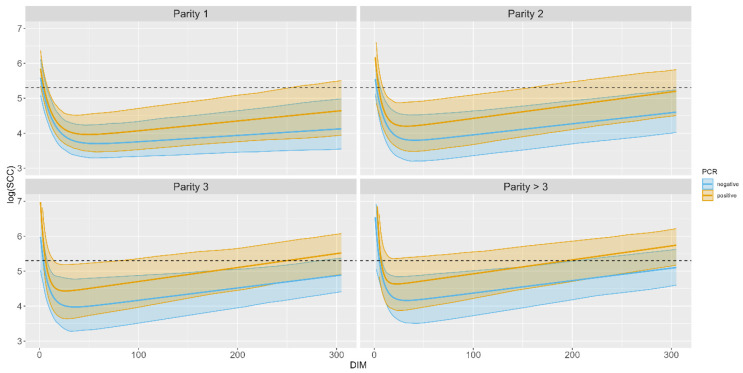
logSCC curves created with means of the four estimated NLME Wilmink parameters plotted for each parity group, separated in PCR positive and PCR negative. The dashed line represents a threshold of 200,000 cells per ml, corresponding to 5.3 on the log scale. The numbers in the grey box refer to the Parity group.

**Table 1 animals-13-02523-t001:** Summary statistics of the somatic cell count (SCC) distribution within the data groups after filtering data limiting SCC to 1 ≤ SCC < 9999. SCC is measured in 10^3^ cells per millilitre of milk.

Parity	PCR	Total SCC Observations	Total Animals	Geometric Mean	Min	Q1	Median	Q3	Max
1	NEG	667,336	77,811	46	1	22	40	83	9978
	POS	420,129	49,107	70	1	28	59	149	9986
2	NEG	381,379	44,210	60	1	26	54	120	9982
	POS	353,818	41,136	103	1	37	95	250	9997
3	NEG	173,335	20,063	77	1	32	70	164	9997
	POS	195,845	22,722	138	1	51	136	347	9998
>3	NEG	104,673	10,351	97	1	38	91	218	9975
	POS	120,395	12,027	171	1	62	172	444	9996

**Table 2 animals-13-02523-t002:** Mean, median values and standard deviation (SD) of the estimated Wilmink-parameters (Param.) fitted with an NLME model.

Param	Parity	PCR	Mean	Median	SD
** *a* **	1	NEG	3.57	3.56	0.229
		POS	3.79	3.78	0.268
	2	NEG	3.64	3.61	0.368
		POS	4.04	4.05	0.387
	3	NEG	3.81	3.80	0.399
		POS	4.31	4.32	0.413
	>3	NEG	4.02	4.02	0.398
		POS	4.52	4.56	0.414
** *b* **	1	NEG	1.82 × 10^−3^	1.70 × 10^−3^	6.98 × 10^−4^
		POS	2.81 × 10^−3^	2.73 × 10^−3^	7.53 × 10^−4^
	2	NEG	3.17 × 10^−3^	3.23 × 10^−3^	6.84 × 10^−4^
		POS	3.79 × 10^−3^	3.79 × 10^−3^	8.04 × 10^−4^
	3	NEG	3.53 × 10^−3^	3.54 × 10^−3^	7.53 × 10^−4^
		POS	3.98 × 10^−3^	3.99 × 10^−3^	9.22 × 10^−4^
	>3	NEG	3.57 × 10^−3^	3.60 × 10^−3^	7.84 × 10^−4^
		POS	3.99 × 10^−3^	4.05 × 10^−3^	9.92 × 10^−4^
** *c* **	1	NEG	−2.54	−2.54	0.114
		POS	−2.43	−2.42	0.214
	2	NEG	−2.25	−2.25	0.137
		POS	−1.93	−1.93	0.310
	3	NEG	−2.03	−2.01	0.293
		POS	−1.62	−1.61	0.148
	>3	NEG	−1.85	−1.85	0.303
		POS	−1.26	−1.25	0.331
** *d* **	1	NEG	2.17	2.18	0.232
		POS	2.24	2.25	0.137
	2	NEG	2.12	2.13	0.143
		POS	2.50	2.47	0.464
	3	NEG	2.49	2.49	0.273
		POS	3.26	3.26	0.0170
	>3	NEG	3.00	3.01	0.532
		POS	5.97	6.03	1.91

**Table 3 animals-13-02523-t003:** Days in milk (DIM) at a minimum of estimated somatic cell count (SCC) curves and the slope based on ΔlogSCC from days in milk (DIM) 100 to 150. The ΔlogSCC slope on the linear part of the logSCC curve suggests how quickly the SCC level increases from DIM 100 to 150.

Parity	PCR	Min logSCC	Min SCC	DIM for Min SCC	ΔlogSCC for DIM Range 100–150
1	NEG	3.700256	40,458	58	0.0018011
	POS	3.954595	52,175	48	0.0028032
2	NEG	3.794039	44,436	40	0.0031664
	POS	4.189145	66,966	31	0.0037942
3	NEG	3.962009	52,563	34	0.0035282
	POS	4.428367	83,794	26	0.0039846
>3	NEG	4.150237	63,449	31	0.0035651
	POS	4.623989	101,900	21	0.0039937

## Data Availability

We do not have the authority to share the data. However, we have provided an R code with simulated data and a pipeline carrying out NLS and NLME model fit with a Wilmink-style function.

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
