# Peer review of "Use of Danish National Somatic Cell Count Data to Assess the Need for Dry-Off Treatment in Holstein Dairy Cattle"

_animals, 2023, doi:10.3390/ani13152523_

Round 1
Reviewer 1 Report
Dear Authors,
Thank you for this paper! To be honest, it takes me time to understand the model that you used to show SCC curves from PCR positive and negative tested (four major mastitis patogens) Danish herds over the entire lactation (divided into parities). I was not able to understand every step you did in detail, but you clearly presented the data with all limitations you find in your model. The results are clear and that was, what I expected when I started to read your paper. So I have no comments for revision. Thank you.
Author Response
Reviewer 1
Thank you for this paper! To be honest, it takes me time to understand the model that you used to show SCC curves from PCR positive and negative tested (four major mastitis patogens) Danish herds over the entire lactation (divided into parities). I was not able to understand every step you did in detail, but you clearly presented the data with all limitations you find in your model. The results are clear and that was, what I expected when I started to read your paper. So I have no comments for revision. Thank you.
AU: Thank you for this comment. This should obviously be clear. We have added an overarching description in the beginning of Section 2 to capture this.
Reviewer 2 Report
The manuscript entitled “Use of Danish national somatic cell count data to assess the need for dry-off treatment in dairy cattle” summarized the Holstein cows in Danish conventional dairy herds were analysed using a nonlinear mixed-effect model with a modified four-parameter Wilmink function. Generally, the manuscript is well written. This paper has several weaknesses and needs improvement before publication.
Comments:
1. Are there any limitations to using this technology? Why did the authors choose this technology instead of other GWAS? Please provide the reason.
2. How much RNA was used in PCR? Was it one-step or 2-step PCR?
3. The abstract is not particularly informative and would benefit from more background.
4. Summarize the abstract, focus on the main findings and mention the small conclusion in at the end of abstract
5. In the Introduction focus on the objectives and insert a few new reference and relevant findings
6. In material and method sections, references are missing.
7. Most of the references mentioned are old and I suggest adding recent references, and the manuscript should be edited accordingly.
8. I suggest the cite following paper in introduction part For more information you can read below reference
H R Jia and others, Lnc-TRTMFS promotes milk fat synthesis via the miR-132x/RAI14/mTOR pathway in BMECs, Journal of Animal Science, 2023;, skad218, https://doi.org/10.1093/jas/skad218
9. Material and method needs to clarifying and summarizing- some detailed needs
The subtitles in the material and method needs to summarizing Ethical approval and references must be mentioned in M&M
10. In the result section, the results were written very poorly. It should be written again and try to avoid a brief introduction in the starting of every result.
This manuscript has major language problems. There are too many for me to modify them all. Authors are strongly encouraged to seek a native English speaker who may assist you modifying the document.
Author Response
Reviewer 2
The manuscript entitled “Use of Danish national somatic cell count data to assess the need for dry-off treatment in dairy cattle” summarized the Holstein cows in Danish conventional dairy herds were analysed using a nonlinear mixed-effect model with a modified four-parameter Wilmink function. Generally, the manuscript is well written. This paper has several weaknesses and needs improvement before publication.
Comments:
- Are there any limitations to using this technology? Why did the authors choose this technology instead of other GWAS? Please provide the reason.
AU: The data were provided to us as are. They are routinely collected data in all Danish dairy herds. GWAS data are not available for mastitis, so we cannot use those.
And yes, the limitations are listed in the last paragraph of the Discussion. If you mean to the SCC-recordings: These have been used for decades in dairy cattle management and are widely known. It is a study on how the somatic cell counts appear across lactation for different lactations and for those with and without detected major mastitis pathogens. We have clarified this as listed for Reviewer 1.
- How much RNA was used in PCR? Was it one-step or 2-step PCR?
AU: We did not use RNA. All routinely collected samples were tested using a commercial kit, which detects DNA. This is a two-step procedure, first with DNA extraction and then detecting of the specific pathogens. We have added a reference.
- The abstract is not particularly informative and would benefit from more background.
AU: We are limited by the word-count, and therefore cannot add more
- Summarize the abstract, focus on the main findings and mention the small conclusion in at the end of abstract
AU: See above
- In the Introduction focus on the objectives and insert a few new reference and relevant findings
AU: It is not clear what the intentions are. We already have 29 references in the Introduction, and these are linked to findings and challenges in relation to the objectives.
- In material and method sections, references are missing.
AU: It is not clear which references that are missing. We have added one for the PCR-method. The rest seems to be covered.
- Most of the references mentioned are old and I suggest adding recent references, and the manuscript should be edited accordingly.
AU: We have reviewed the references and have used those that highlights the challenges. It is not clear which ones that might be outdated. We have new and old. We do not notice any that are outdated.
- I suggest the cite following paper in introduction part For more information you can read below reference
H R Jia and others, Lnc-TRTMFS promotes milk fat synthesis via the miR-132x/RAI14/mTOR pathway in BMECs, Journal of Animal Science, 2023;, skad218, https://doi.org/10.1093/jas/skad218
AU: We do not find this paper relevant for our study. This is not about milk fat synthesis.
- Material and method needs to clarifying and summarizing- some detailed needs
The subtitles in the material and method needs to summarizing Ethical approval and references must be mentioned in M&M
AU: The intention with this is not clear. Furthermore, this is a database study, and ethical approval not required according to Danish legislation.
- In the result section, the results were written very poorly. It should be written again and try to avoid a brief introduction in the starting of every result.
This manuscript has major language problems. There are too many for me to modify them all. Authors are strongly encouraged to seek a native English speaker who may assist you modifying the document.
AU: Proofreading by a professional proof-reading company has been done. Furthermore, one of the authors (M. Denwood) is a native English speaker. We have modified according to the proof-reading.
Reviewer 3 Report
The paper is titled "Use of Danish national somatic cell count data to assess the need for dry-off treatment in dairy cattle." This study is an interesting observational retrospective study that aims to characterize the association between PCR test results at dry-off and SCC-curves before dry-off. Specifically, it aims to fit an SCC-curve model that can be used to support treatment decisions for four major IMI pathogens.
I believe the article is well-written and provides good statistics based on a large amount of analyzed data. However, I think there is untapped potential in this study that is not adequately expressed, probably to be studied in a deeper way in future papers. Particularly, it is not clear how the obtained results can be useful for selecting treatment options for IMI pathogens. In addition, you could also focus on how the model used and the PCR-SCC could be more valuable in providing accurate diagnosis and treatment for IMI, to possibly establish new and more effective rules for large-scale control.
INTRODUCTION:
Line 74: “However, this general threshold ignores important factors affecting SCC” please specify better what do you mean.
DISCUSSION:
Please find comment above in general part.
Lines 315-318. Since it is stated that “is reasonable to assume that the 200,000 cells/ml limit should not be applied before 40 DIM”, we suggest that you indicate an alternative or supplemental approach to obtain reliable results for intervention, to be applied before 40 DIM.
Line 357. Dot missing after “animals”.
Author Response
Reviewer 3
The paper is titled "Use of Danish national somatic cell count data to assess the need for dry-off treatment in dairy cattle." This study is an interesting observational retrospective study that aims to characterize the association between PCR test results at dry-off and SCC-curves before dry-off. Specifically, it aims to fit an SCC-curve model that can be used to support treatment decisions for four major IMI pathogens.
I believe the article is well-written and provides good statistics based on a large amount of analyzed data. However, I think there is untapped potential in this study that is not adequately expressed, probably to be studied in a deeper way in future papers. Particularly, it is not clear how the obtained results can be useful for selecting treatment options for IMI pathogens. In addition, you could also focus on how the model used and the PCR-SCC could be more valuable in providing accurate diagnosis and treatment for IMI, to possibly establish new and more effective rules for large-scale control.
AU: Thank you for these suggestions. We were clearly missing the perspectives and have added a paragraph in the end of the Discussion. This also addresses the request for cow-level treatment decisions. This is a topic for our subsequent paper, and at this point not feasible to also include in this paper.
INTRODUCTION:
Line 74: “However, this general threshold ignores important factors affecting SCC” please specify better what do you mean.
AU: We have added a sentence to be more explicit.
DISCUSSION:
Please find comment above in general part.
Lines 315-318. Since it is stated that “is reasonable to assume that the 200,000 cells/ml limit should not be applied before 40 DIM”, we suggest that you indicate an alternative or supplemental approach to obtain reliable results for intervention, to be applied before 40 DIM.
AU: We have added an alternative.
Line 357. Dot missing after “animals”.
AU: This dot has been added.
Reviewer 4 Report
The manuscript authored by Maj Beldring Henningsen et al., describes the use of Danish national somatic cell count data to assess the need for dry-off treatment in dairy cattle. The objective was to compare somatic cell count curves throughout lactation in dairy cows with PCR test results for four major intramammary infection pathogens. The authors focus on the Danish dairy cattle database, selecting Holstein cattle from conventional farms that were tested for PCR in the post-lactational phase. In fact, it is an observational study of a retrospective registry that includes 10 years of data taken from different farms in the country on animals with 1, 2, 3, or more than 3 parities.
From a purely numerical and statistical point of view, the results show that the number of somatic cells increases with the number of calvings and in those animals that are PCR positive. This is not a novelty in itself as it has been demonstrated previously. However, some important biological aspects have not been taken into account, as described below:
The title of the paper refers to "dairy cattle". This assumption should include all cattle destined for milk production. However, only the Holstein breed is included in the document. Actually, more than 550,000 animals (organic herds and other breeds) have been excluded from the study. What was the reason for the exclusion? I believe that it would have been interesting to have this data to compare with the current data presented.
The authors also describe several limitations of the study among which they note “Another limitation needing to be highlighted is since register data are not collected by the researcher [50] important information on the herds may be missing in the study”. In this way, it would have been interesting to know how the animal data have progressed over time, taking into account that in 10 years, sanitary management, feeding, and milking patterns, among others, have changed.
For the above reasons, I believe that modifications should be made based on the comments made to improve the quality of the work.
Moderate editing of English language required
Author Response
Reviewer 4
The manuscript authored by Maj Beldring Henningsen et al., describes the use of Danish national somatic cell count data to assess the need for dry-off treatment in dairy cattle. The objective was to compare somatic cell count curves throughout lactation in dairy cows with PCR test results for four major intramammary infection pathogens. The authors focus on the Danish dairy cattle database, selecting Holstein cattle from conventional farms that were tested for PCR in the post-lactational phase. In fact, it is an observational study of a retrospective registry that includes 10 years of data taken from different farms in the country on animals with 1, 2, 3, or more than 3 parities.
From a purely numerical and statistical point of view, the results show that the number of somatic cells increases with the number of calvings and in those animals that are PCR positive. This is not a novelty in itself as it has been demonstrated previously. However, some important biological aspects have not been taken into account, as described below:
The title of the paper refers to "dairy cattle". This assumption should include all cattle destined for milk production. However, only the Holstein breed is included in the document. Actually, more than 550,000 animals (organic herds and other breeds) have been excluded from the study. What was the reason for the exclusion? I believe that it would have been interesting to have this data to compare with the current data presented.
AU: Good point. We are providing a proof-of-concept, and therefore non-Holsteins were excluded. This information has been added. Due to differences in legislation on treatment for organic and conventional herds, the epidemiology can likely be different. Therefore, organic herds were also excluded. We have added “Holstein” to the title to reflect the above.
The authors also describe several limitations of the study among which they note “Another limitation needing to be highlighted is since register data are not collected by the researcher [50] important information on the herds may be missing in the study”. In this way, it would have been interesting to know how the animal data have progressed over time, taking into account that in 10 years, sanitary management, feeding, and milking patterns, among others, have changed.
AU: We do not have this kind of information, and therefore it is not possible to include.
For the above reasons, I believe that modifications should be made based on the comments made to improve the quality of the work.
Moderate editing of English language required
AU: See comment above for Reviewer 2.